# Bandwidth Improvement in Bow-Tie Microstrip Antennas: The Effect of Substrate Type and Design Dimensions

**Halgurd N. Awl [1], Yadgar I. Abdulkarim [2,3] , Lianwen Deng [2], Mehmet Bakır [4] , Fahmi F. Muhammadsharif [5] , Muharrem Karaaslan [6], Emin Unal [6] and Heng Luo [2,*]**

[1] Department of Communication, Engineering College, Sulaimani Polytechnic University, Sulaimani 46001, Iraq; halgurd.awl@spu.edu.iq

[2] School of Physics and Electronics, Central South University, Changsha, Hunan 410083, China; yadgar.kharkov@gmail.com (Y.I.A.); denglw@csu.edu.cn (L.D.)

[3] Physics Department, College of Science, University of Sulaimani, Sulaimani 46001, Iraq

[4] Department of Computer Engineering, Yozgat Bozok University, Yozgat 66900, Turkey; mehmet.bak@gmail.com

[5] Department of Physics, Faculty of Science and Health, Koya University, Koya 44023, Iraq; Fahmi982@gmail.com

[6] Department of Electrical and Electronics, Iskenderun Technical University, Hatay 31100, Turkey; muharrem.karaaslan@iste.edu.tr (M.K.); emin.unal@iste.edu.tr (E.U.)

* Correspondence: luohengcsu@csu.edu.cn

**Abstract:** In this work, the impact of substrate type and design dimensions on bow-tie microstrip antenna performance and bandwidth improvement is presented both numerically and experimentally at 4–8 GHz. The finite integration technique (FIT)-based high-frequency electromagnetic solver, CST Microwave Studio, was used for the simulation analysis. For this purpose, four bow-tie microstrip antennas were designed, fabricated, and measured upon using different materials and substrate thicknesses. Precise results were achieved and the simulated and experimental results showed a good agreement. The performance of each antenna was analyzed and the impact of changing material permittivity, antenna dimensions and substrate thicknesses on antenna performance were investigated and discussed. The measured results indicated that the slot bow-tie antenna, which is one of the novel aspects of this study, is well matched and a 2-GHz bandwidth [5–7 GHz] is obtained, which is about 50% bandwidth in comparison with the wideband applications [4–8 GHz]. The proposed structure is useful in ultra-wideband (UWB) applications. This study provides guidance in selecting material types and thicknesses for microstrip antennas based on desired applications.

**Keywords:** bow-tie antenna; bandwidth; materials permittivity; substrate effect; antenna performance

## 1. Introduction

Microstrip antennas with a wide operation bandwidth have received much attention due to their employment in different applications with easy tuning options [1–5]. Progress in handheld devices requires these devices to be thinner to support small-sized appliances with improved performance [6–8]. In this way, antennas with different shapes have been studied to achieve miniaturization and bandwidth enhancement, such as circle [9], ellipse [10], triangle [11], fractal [12], U-shaped [13], etc. [14–16].

Bow-tie antenna, which is a type of microstrip antenna, has been used in different applications. A ground penetrating radar (GPR) was recently designed by Karamzadeh et al. in 2016 [17], in which a bow-tie antenna was utilized aiming at improving its performance. Also, Liu et al. developed

a bow-tie antenna that was used in GPR applications [18]. They adjusted the parameters of the antenna to change its equivalent capacitance and inductance, while it was tuned to operate in the frequency range from 0.5 to 1.2 GHz, thereby realizing an ultra-wideband antenna. Very recently, Sahoo and Vakola proposed a novel wideband cylindrical conformal antenna for global positioning system (GPS) applications by using a wide slot in the ground plane, incorporating a bow-tie-shaped patch antenna [19]. The effect of graphene conductive ink on the performance of bow-tie-shaped antenna was also studied by Ram et al. [20]. It was seen that the developed design is well compatible for applications in LTE, WiMAX and Wi-Fi. In 2019, another bow-tie antenna was designed by Bozdag and Secmen for entire band application of GPS (L5), PCS, IMT-2000, Bluetooth, Wi-Fi, WiMAX bands, and part of ultra-wide band (UWB) frequency range [21]. In order to utilize bow-tie type UWB antennas Kim et al. in [22] designed a balun which is integrated into the antenna.

In the current research work, three different bow-tie-shaped microstrip antennas are designed, fabricated and experimentally tested. The first and second batch of the devices is fabricated on different thicknesses of RT 5880 substrate. Simulations and experimental results of the first batch showed that the antenna is well matched and the S11 is about 37 dB. The antenna is a single band operating at 7 GHz with 0.7 GHz (6.7 GHz–7.4 GHz) impedance bandwidth. The second batch is also printed on RT5880 materials with different substrate thickness. It was noticed that higher thickness provided relatively wider bandwidth, which is around 0.9 GHz (6.7 GHz–7.6 GHz), and the percentage bandwidth was increased from 10% to 12.5%. The third design is suggested to show a broadband application of bow-tie-shaped antenna. In the third batch, 1.4 GHz bandwidth was obtained. All of the three designs are under the UWB band and can be used for different microwave applications. From the results of the paper, it was found that the smaller thickness provides wider bandwidth with lower gain. Therefore, the thickness and relative permittivity of the substrate can be selected based on the application. It means the substrate thickness and type can be used to control obtained gain and bandwidth. This work can be used as a practical guide that can help researchers to view substrate selection as a method to control antenna performance (gain, bandwidth, polarization). This point can be considered as the novelty of this work, which may not have been seen before.

As is generally known, the design of a slotted bow-tie antenna depends on the applications and if it is working in frequency or time domain. In frequency-domain applications, slots are often used in a bow-tie antenna to increase the antenna bandwidth [23–25]. In this case, two parameters are of importance—namely the slot width and location. The slot dimensions, which is another novel side of this study, are chosen especially for the required bandwidth and gain.

The rest of this paper is organized as follows: In Section 2, the design of bow-tie-shaped microstrip antennas is presented, while the fabrication process of the antennas are presented in Section 3. Further, the simulation and experimental results along with discussion are given in Section 4, and the main conclusions are drawn in Section 5.

## 2. Design of Bow-Tie Microstrip Antennas

The proposed bow-tie microstrip structure, as shown in Figure 1, is presented to improve the antenna performance. The CST Microwave Studio program is used as a main program to design and simulate the antennas. AutoSketch software, which is an inclusive electrical structure drawing software, is used to draw the designed antennas with 1:1 scale in order to print the desired antenna on laser paper for fabrication. The purpose of this program is to avoid any error and mistakes in the scale of the fabricated structures.

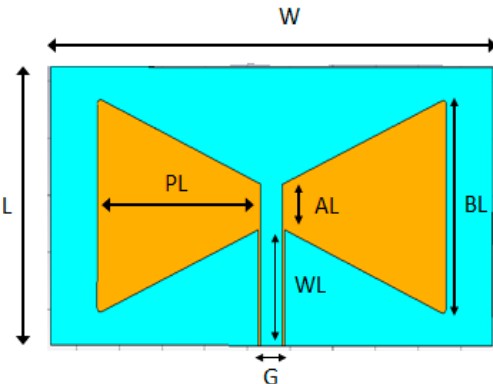

**Figure 1.** The design geometry of antenna A and B with same RT/Duroid 5880 materials substrate and different substrate thickness.

Material selections are one of the important issues that should be carefully considered during any design process. The material thicknesses can also have a crucial influence on the microstrip antenna design, hence it should be precisely selected. In the design of the bow-tie microstrip structure, RT/Duroid 5880 and RT/Duroid 5880 LZ, from the Regress Company, are used as raw materials. These materials are chosen because they are cheap and easily obtained. They are available under different substrate thicknesses. The cladding layer of these materials is copper with constant thickness of 35 micrometres. The same material type and thickness are modelled and used in the CST simulation.

*2.1. Antenna Design by Using RT/Duroid 5880 Material*

In this section, two bow-tie antennas are designed with the same material and different thicknesses. The designed antennas are fabricated and tested. The measured results are compared to the simulation results, while results of both antennas are compared and analyzed. In order to avoid confusion and to easily distinguish between the two antenna results, the capital letter A and B are respectively assigned for the first and second antenna. The antenna A, which is shown in Figure 1 is designed and simulated by CST Microwave Studio. The relative permittivity of the material is 2.2 and the substrate thickness is 1.57 mm. The two microstrip lines are used to excite the antenna with 50 Ω input impedance waveguide port. The two triangular shapes are symmetric (exactly the same). Table 1 shows the dimensions of antenna A in millimeters, also Figure 1 shows the antenna B geometry, which is also fed by the same way of antenna A using two microstrip lines connected to the waveguide port. Antenna B has a larger substrate thickness of 3.175 mm. The dimensions of antenna B are presented in Table 2.

**Table 1.** Antenna A parameters.

| Antenna Parameters | Dimensions (mm) |
| --- | --- |
| Patch length PL | 35 |
| Base length BL | 33 |
| Gap | 4 |
| Feed line width Wf | 0.6 |
| Apex length AL | 5.5 |
| Substrate length L | 43.35 |
| Substrate width W | 87 |

**Table 2.** Antenna B Parameters.

| Antenna Parameters | Dimensions (mm) |
|---|---|
| patch length PL | 35.5 |
| Base length BL | 30.5 |
| Gap | 3.7 |
| Feed line width Wf | 0.5 |
| Apex length AL | 5.5 |
| Substrate length L | 42 |
| Substrate width W | 85 |

### 2.2. Antenna Design by Using RT/Duroid 5880LZ Material

In the second stage, two different antennas were designed and fabricated using the same materials but different thicknesses. This material has a lower relative permittivity and thicknesses than that used in Section 2.1. Again, to avoid confusion during comparison and analysis of the results, the first antenna is named antenna A while the second one is antenna B.

The purpose of using this material is to investigate the impact of relative permittivity and substrate thicknesses on the antenna performance. Furthermore, design and fabrication of antennas with different relative permittivity and substrate thicknesses gives a practical experience to choose the best material for antenna design and hence achieving more accurate results.

The geometry of antenna A and antenna B are shown in Figure 2. The antenna A established on a 0.508 mm substrate having a relative permittivity of 1.96. The antenna is symmetric and all antenna dimensions are presented in Table 3 (all the dimensions are in millimeters.)

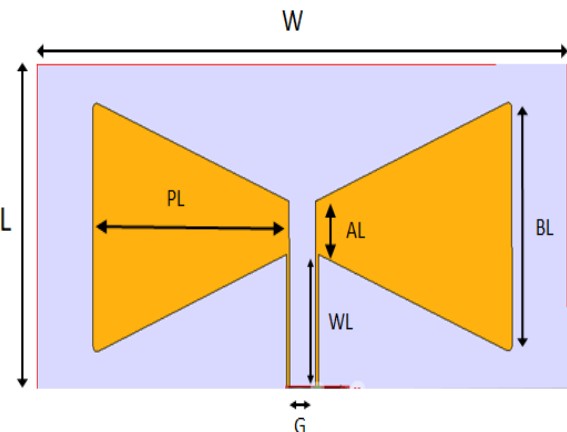

**Figure 2.** Design of the geometry of antenna A and B with same substrate RT/Duroid 5880LZ material and different thickness.

**Table 3.** Antenna A parameters.

| Antenna Parameters | Dimensions (mm) |
|---|---|
| Patch length pL | 36 |
| Base length BL | 34 |
| Gap | 4 |
| Feed line width Wf | 0.5 |
| Apex length AL | 5.5 |
| Substrate length L | 45 |
| Substrate width W | 87 |

Antenna B was built on a 1.27-mm-thick substrate with a relative permittivity of 1.96. Table 4 presents the required dimensions of antenna B.

**Table 4.** Antenna B parameters.

| Antenna Parameters | Dimensions (mm) |
| --- | --- |
| Patch length pL | 36 |
| Base length BL | 34 |
| Gap | 4 |
| Feed line width Wf | 0.5 |
| Apex length AL | 5.5 |
| Substrate length L | 45 |
| Substrate width W | 87 |

### 2.3. Design of a Broadband Bow-Tie-Based Antenna

In this stage, by introducing a small sector stub near the feed line and bending the corners of both triangular shapes, the wideband bow-tie antenna is designed and simulated. This small sector plays an important role in increasing bandwidth because the current is strengthened around the stub, which acts upon improving the electric field distribution. Bending the corners of the stub and presenting both triangular patch shapes are participating in the enhancement of the bandwidth [1]. The antenna was refabricated on different substrate thicknesses in order to optimize its performance and hence obtaining better results. The geometry of the designed antenna is shown in Figure 3, while the dimensions of the antenna and slots are presented in Table 5.

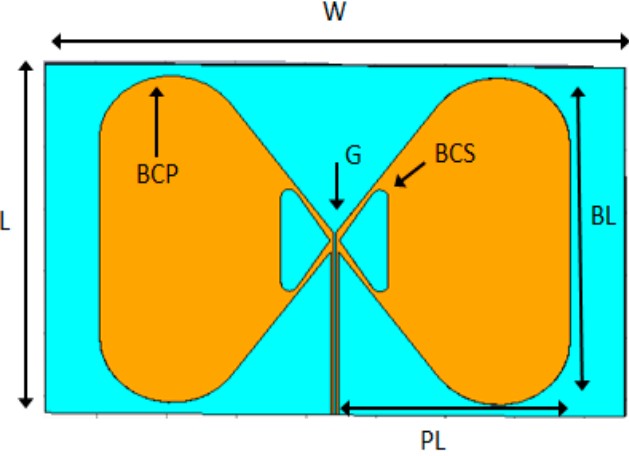

**Figure 3.** The geometry of the proposed broadband antenna.

**Table 5.** The antenna and slot diminutions.

| Antenna Parameters | Description | Values (mm) |
| --- | --- | --- |
| W | Antenna width | 54 |
| L | Antenna length | 80 |
| BL | Triangular patch length | 93 |
| PL | Base length of the triangle | 32.5 |
| G | Gap | 0.3 |
| BCP | Bending radius of the patch corners | 10 |
| BCS | Bending radius of the slot corners | 1.15 |
| Bs | Base length of the slot | 21.44 |
| Ls | Length of the slot | 7.4 |
| Wf | Width of the feed lines | 0.38 |
| Apex | Apex length of triangular top | 2.35 |

### 3. Fabrication of the Proposed Bow-Tie Microstrip Antenna

The proposed bow-tie antenna for UWB applications has been fabricated as illustrated in Figure 4. Basically, the antennas are designed to be operated at 7 GHz, which is relatively the center of the UWB frequency band. However, the designs can be easily modified in order to tune the operation frequency for various UWB applications. The materials used in the fabrication of bow-tie microstrip are RT/Duroid 5880 and RT/Duroid 5880, while considering different thicknesses. In the first stage of the fabrication process, RT/Duroid 5880 was used in two different thicknesses (1.57 mm and 3.175 mm) with a relative permittivity of about 2.2. In the design of both antennas, the two microstrip lines were used to excite the antenna with 50 Ω input impedance waveguide port. The two triangular shapes were exactly similar, while their substrate thickness was different.

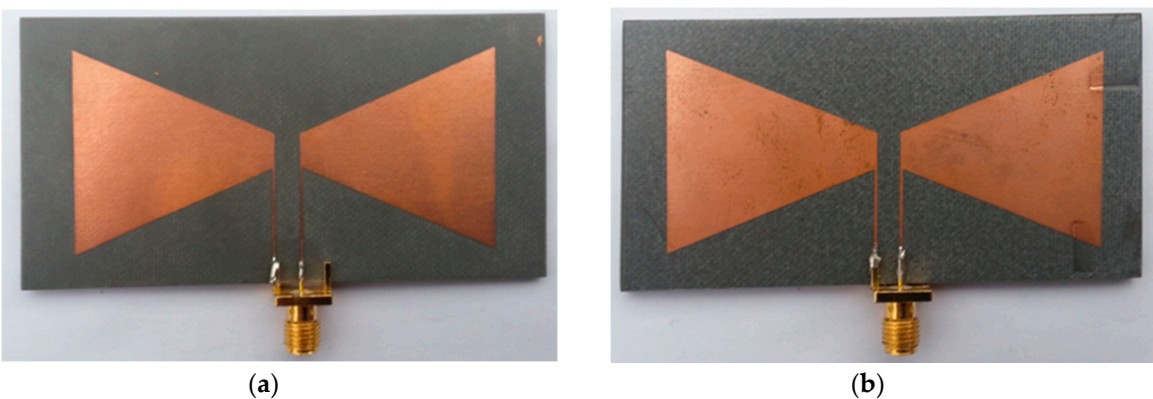

(**a**)　　　　　　　　　　　　　　　　　　　　　　　　(**b**)

**Figure 4.** Fabricated bow-tie (**a**) antenna A with a thickness of 1.57 mm and 0.508 mm, (**b**) antenna B with a thickness of 3.175 mm and 1.27 mm.

The fabricated antennas with 50 Ω SMA (Sub-Miniature version A) connectors are shown in Figure 4. The positive pin of the SMA connecter is soldered with the right triangular patch, which behaves as a positive terminal. While the left triangular patch is connected to the ground of an SMA connecter and it behaves as a negative junction. The dimensions of both fabricated antennas are shown in Tables 1 and 2, respectively.

In addition, two more antennas were designed by using RT/Duroid 5880 LZ material with different thicknesses. Both antennas have a relative permittivity of 1.96 with a substrate thickness of 0.508 mm and 1.27 mm, respectively. The required dimensions for the fabrication process are shown in Tables 3 and 4, respectively, while the physical presentation of the antennas with SMA connectors are shown in Figure 4.

The proposed wideband bow-tie antenna has been fabricated two times with the same materials but in different thicknesses. For more precise results, the width and length of the antenna were considered to be 54 mm and 80 mm by using an E33 model LPKF stands for "Leiterplatten-Kopierfräsen", which when translated means "circuit board copy milling" prototyping printed circuit board (PCB) machine. This is where a single-sided substrate covered with copper of a constant thickness of 35 micrometers was used. The thicknesses of both fabricated antennas were 0.508 mm and 1.27 mm, respectively. All required dimensions are shown in Table 5. After the fabrication of the antenna, a 50-Ohm SMA connector has been soldered to the feeding line, as illustrated in Figure 5. The experimental study has been achieved via a PNA-L Agilent vector network analyzer (VNA) with an operating frequency ranging from 10 MHz to 43.5 GHz. The VNA was first calibrated by using the proper calibration kit having short circuit, open circuit and load apparatus. The calibration of VNA was realized in the operating frequency range of antenna and this range was between 4 GHz and 8 GHz. Finally, the antenna has been connected to the VNA and the return loss parameter, gain and radiation pattern were measured.

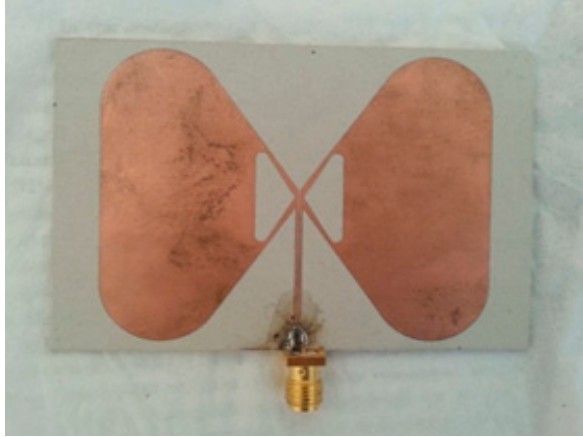

**Figure 5.** Image of the fabricated wideband bow-tie antenna.

## 4. Results and Discussion

The simulated and fabricated results for the antenna incorporating RT/Duroid 5880 materials as the substrate with thickness of 1.57 mm were compared with each other, as shown in Figure 6. The transient solver of the program was used to achieve the desired result. In order to obtain a precise result, the mesh setting was accurately adjusted and the required number of meshes was applied. The network analyzer was used to measure the return loss. It can be seen from Figure 6 that the antenna is well matched and the S11 is about 37 dB. The antenna is a single band operated at 7 GHz with 0.7 GHz (6.7 GHz–7.4 GHz) impedance bandwidth.

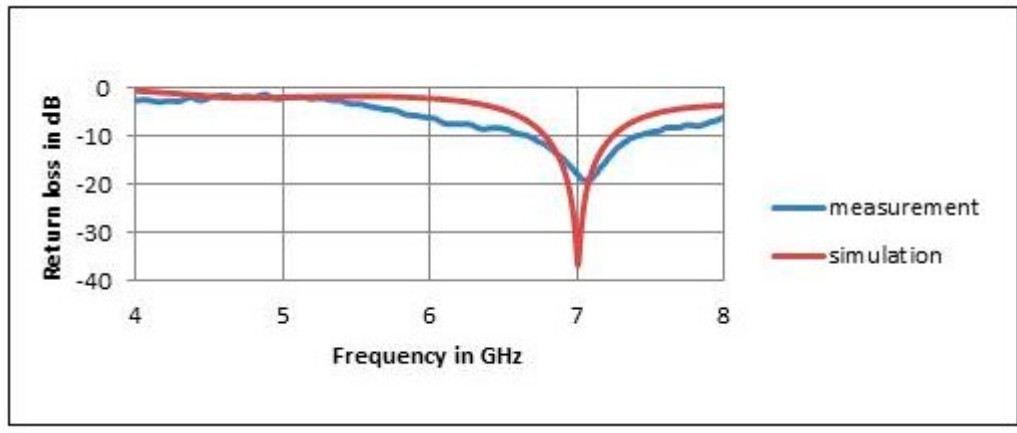

**Figure 6.** The comparison of simulated and measured return loss of antenna A.

Similarly, the simulated and measured results for the proposed antenna including RT/Duroid 5880 materials with different thicknesses are shown in Figure 7. The same steps as of antenna A were repeated in order to achieve the best possible results.

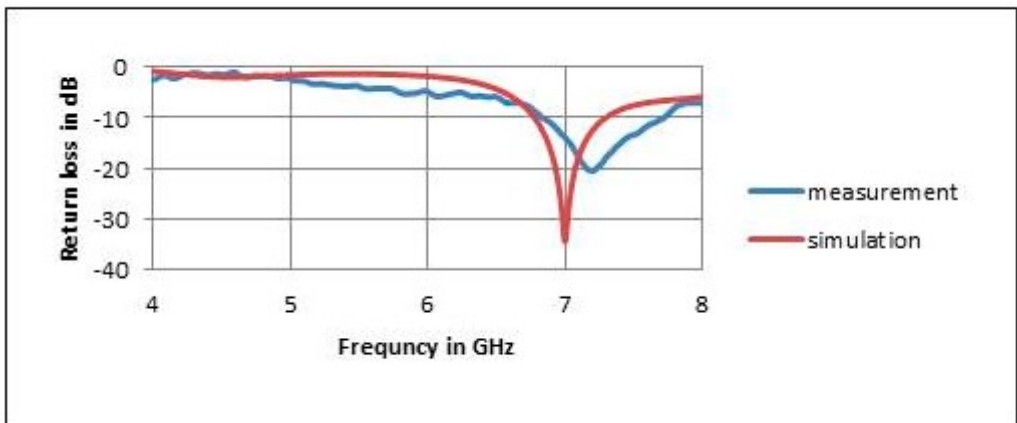

**Figure 7.** The comparison of simulated and measured return loss of antenna B.

One can see from Figure 7 that the simulation result is in accordance with the measured result and the antenna is well matched. However, a small shift can be seen in the operation frequency of the antenna. This might have occurred due to a small scratch in the radiated patch, which occurred during the fabrication process. However, the second antenna, which has is less thick, provided a relatively wider bandwidth, which is around 0.9 GHz (6.7 GHz–7.6 GHz), and the percentage bandwidth is 12.5%, while the percentage bandwidth for antenna A was approximately 10%.

The far-field radiation pattern of the antennas could not be measured due to the unavailability of proper device and tools to perform this process. Therefore, only the simulated E plane and H plane radiation pattern of the two antennas are presented in Figure 8. It can be seen from the figure that both antennas provide directional radiation pattern in both planes, while the antenna has a higher side lobe level in E plane of both antennas with a different main lob direction. Antenna B provided a lower side lobe level in the H plane with wider angular of the main lobe compared to that of antenna A.

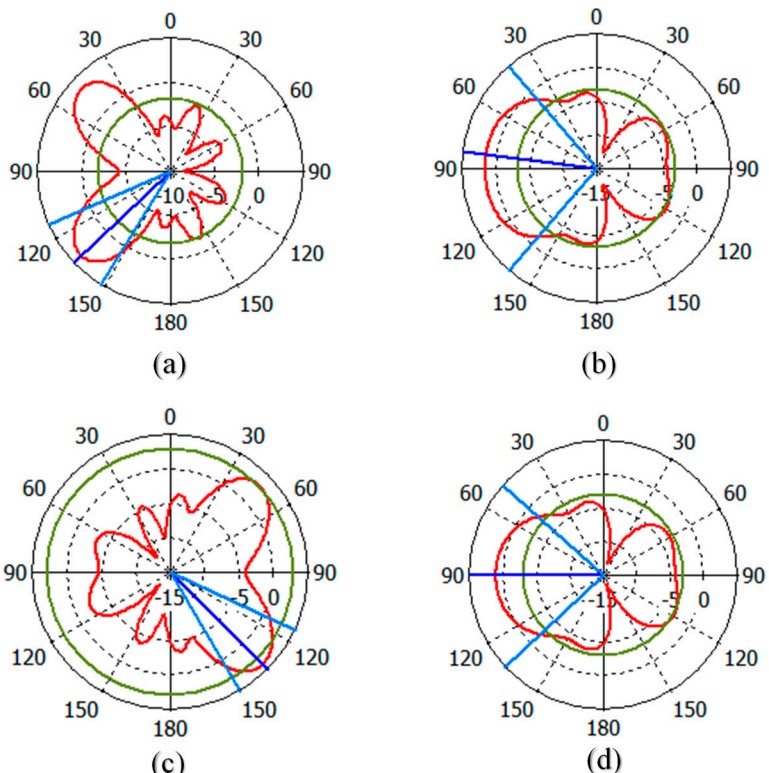

**Figure 8.** The radiation pattern of the proposed antennas (**a**): E plane of antenna A (**b**), H plane of antenna A (**c**), E plane of antenna B (**d**), and H plane of antenna B (**b**).

In the simulated design and fabricated antenna using RT/Duroid 5880 LZ substrate material with a thickness 0.508 mm and relative permittivity of 1.96, the simulated and measured results for the return loss (S$_{11}$ parameters) were recorded, as illustrated in Figure 9.

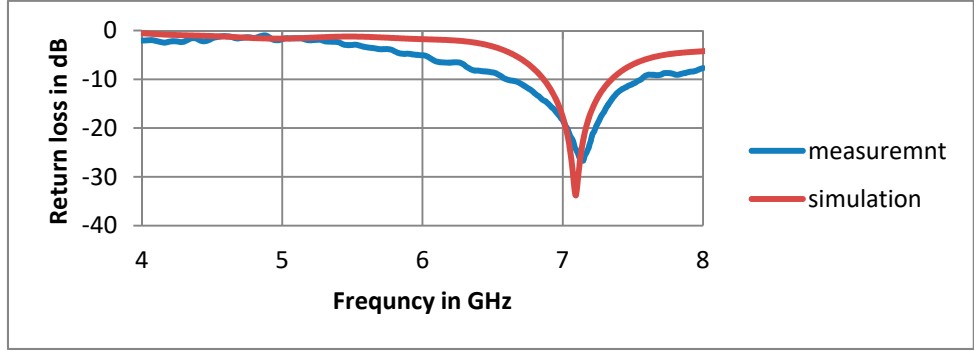

**Figure 9.** The simulated and measured return loss of antenna A with thickness of 0.508 mm.

Figure 9 shows the simulated and measured results for the return loss (S$_{11}$ parameters) when the substrate is yet RT/Duroid 5880 LZ but with thickness of 1.27 mm and same relative permittivity of 1.96.

It can be seen from the Figure 10 that both antennas are well matched and the percentage bandwidths of both antennas are almost the same which are 0.13% in both cases (second antenna gives 0.9 GHz (6.5 GHz–7.4 GHz) and first antenna gives 0.92 GHz (6.6 GHz–7.52 GHz)). Antenna B should have been provided a relatively wider bandwidth than that of the antenna A, but this difference was not apparent in the measured result; this could be due to the (SAM) Sub-Miniature version A connection and fabrication tolerances. In addition, it can be observed that this material ($\varepsilon_r$ = 1.96) provides better matching and wider bandwidth compared to the RT/Duroid 5880 materials.

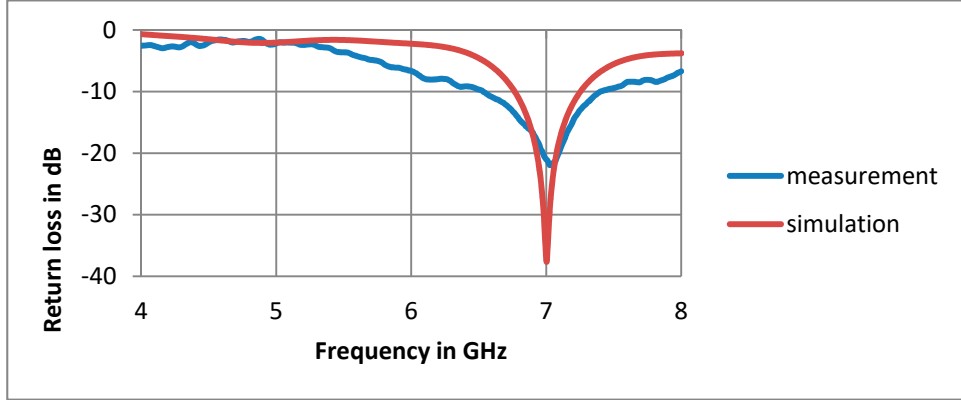

**Figure 10.** The simulated and measured return loss of antenna B with thickness of 1.27 mm.

The far-field pattern of both antennas is presented in Figure 11. The shape of radiation patterns of antenna A and B in both planes are approximately the same. Whereas, the antenna A provide smaller 3 dB angular width of the main lobe with lower side lobe level in comparison with antenna B. Therefore, it can be said that, changing the material permittivity and substrate thickness have a small effect on the radiation a pattern properties.

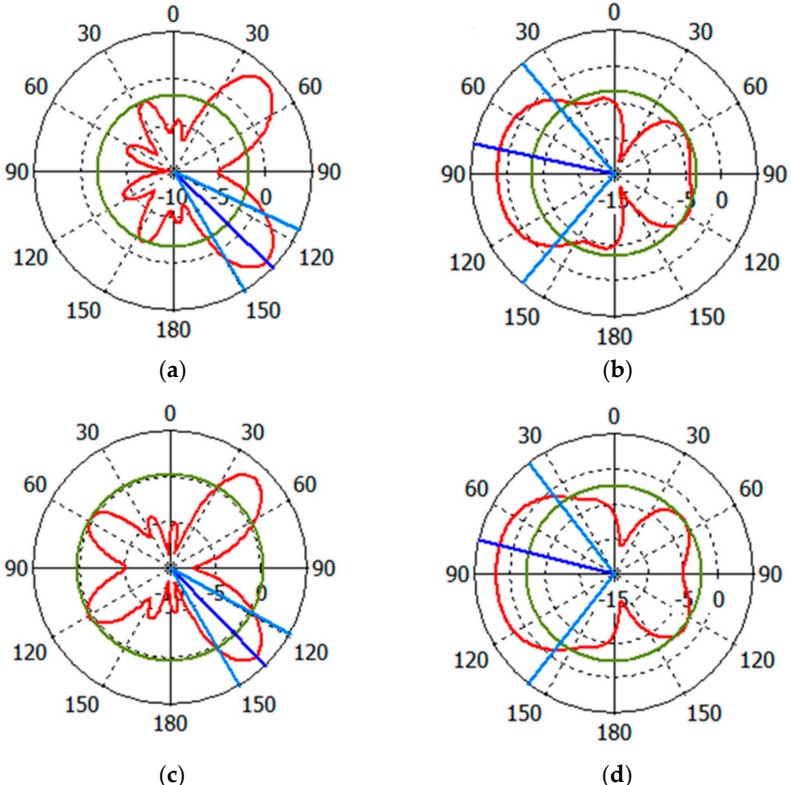

**Figure 11.** The radiation pattern of the proposed antennas (**a**), E plane of antenna A (**b**), H plane of antenna A (**c**), E plane of antenna B (**d**) and H plane of antenna B.

The realized gain over operating frequency range of the four antennas is plotted in the Figure 12. As can be seen from the figure, the antenna with lower relative permittivity of 1.96 provided higher gains, while when the substrate thickness is increased to 3.175, the gain also increased. Therefore, it can be said that the realized gain is inversely proportional with relative permittivity and directly proportional with substrate thickness.

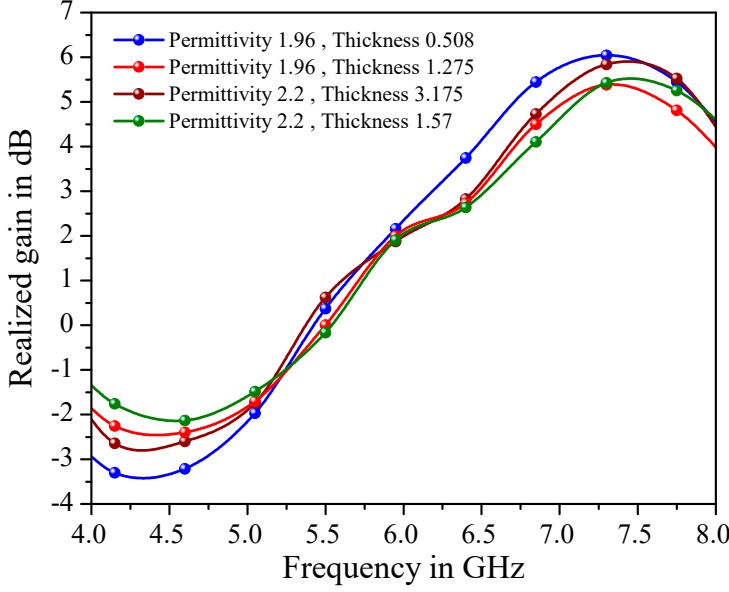

**Figure 12.** Realized gain of the proposed antennas versus the frequency in the range of interest.

Finally, the polarization of the four antennas was monitored. For this purpose, the axial ration of the antennas was plotted in Figure 13. Based on the 3 dB criteria, all antennas provided circular polarization, while an improved circular polarization was obtained from the antennas with relative permittivity of 1.96 and 2.2 with a substrate thickness of 1.275 and 3.175, respectively.

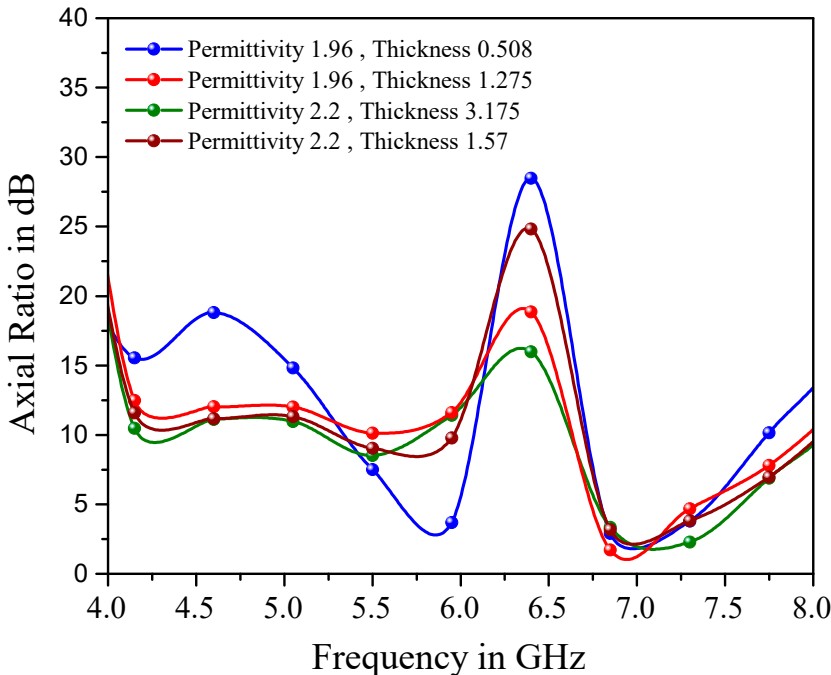

**Figure 13.** Simulated axial ratios of the antennas over the frequency range of interest.

As demonstrated in the previous sections, the proposed antennas have a relatively narrow bandwidth. Therefore, broadband bow-tie slot antenna was designed, fabricated and tested. The measured result is compared with the simulation result. The fabricated antenna is measure by the VNA network analyzer. The network analyzer is calibrated by using a calibration tool in order to get more precise results, such as open, short and broadband loads. The simulated and measured return losses are plotted in Figure 14.

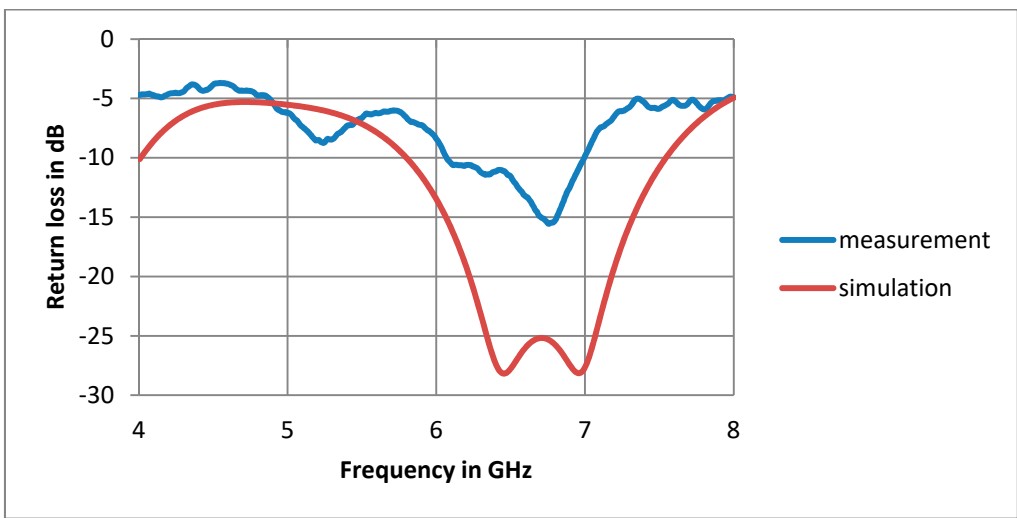

**Figure 14.** The simulated and measured return losses of the slot antenna (thickness = 1.27 mm).

From Figure 14, it can be observed that the measured result is relatively matched with the simulation result. The simulation result has a broad bandwidth which is about 33% with −27 dB return loss, while the measured result shows the smaller bandwidth of 14% and the mismatch was higher. The inaccurate result may be referring to the soldering of the SMA connecter. Because the antenna gab, which is located between the two feeding lines, is very narrow, as mentioned in Table 5, it is difficult to connect the SMA exactly in the right location. The fabricated antenna with an SMA connecter is shown in Figure 6.

For the purpose of getting better results and improving the amount of mismatching, the smaller substrate, which is 0.508 mm, was used. The antenna was fabricated with this smaller substrate in which all the other parameters were kept constant as previously illustrated in Table 5. The fabricated antenna was tested with the VNA analyzer again. The simulated and measured results are plotted in Figure 15.

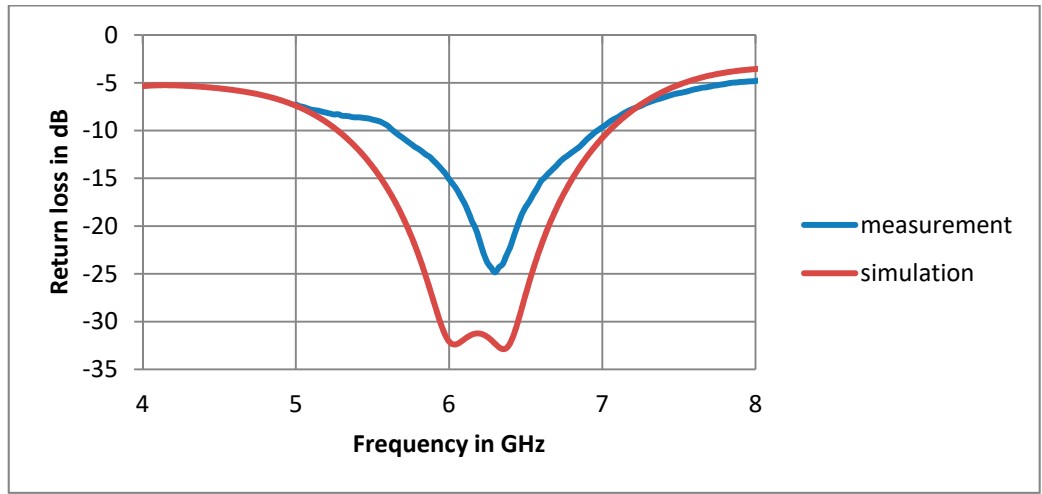

**Figure 15.** The simulated and measured return losses of the slot antenna (thickness = 0.508 mm).

From Figure 15, it can be noted that more accurate result was obtained because the measured result is highly correlated with the simulation result. The measured return loss is around −25 dB and the obtained bandwidth is about 1.4 GHz, which is about 22% (obtained bandwidth is 1.3 GHz with 5.65 GHz lower and 6.95 GHz higher frequency limits).

Again, due to the availability of the devices, the far-field pattern is not measured. Therefore, only the simulated far-field pattern is presented in Figure 16.

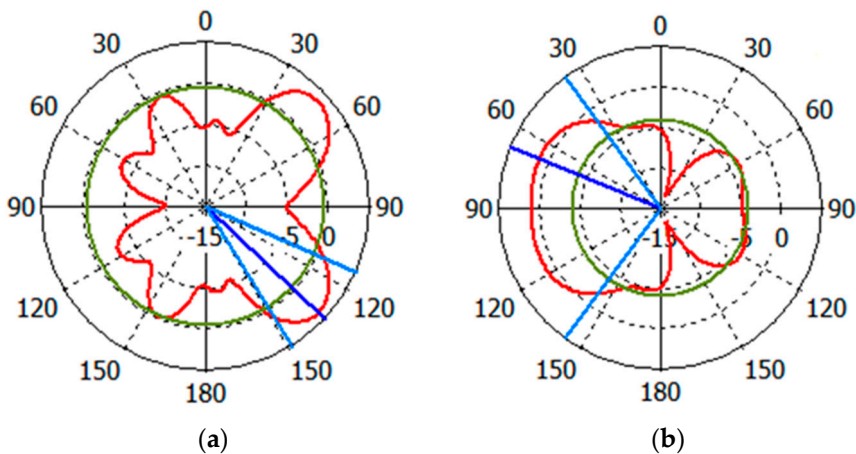

(a)  (b)

**Figure 16.** The radiation pattern of the bow-tie slot antenna (**a**), E-plane and H-plane (**b**).

## 5. Conclusions

In summary, various bow-tie microstrip antennas were designed and verified experimentally to support the optical antenna in the range of 4–8 GHz for wideband (UWB) applications. The finite integration technique (FIT)-based high-frequency electromagnetic solver, CST Microwave Studio, was used for numerical analysis. The designed antennas were fabricated and tested for different materials and substrate thicknesses. It has been observed that all antennas are well matched and accurate results are achieved. In order to overcome the bandwidth limitation of these antennas, broadband slot bow-tie antenna has been introduced. The slot bow-tie antenna was modelled, designed and fabricated. The obtained percentage bandwidth is 20% with a return loss of −25 dB. However, the measured result shows that the bow-tie slot antenna is not perfectly matched, as was expected. Therefore, the slot bow-tie antenna was fabricated again with the lowest available relative permittivity of 1.96 and substrate thickness of 0.508 mm. As expected, the measured result indicated that the slot bow-tie antenna is well matched, and the obtained percentage bandwidth is about 22% with a return loss of −25 dB. As briefly explained here, the designed slotted bow-tie antenna, which is shown in Figures 3 and 5, and experimentally verified as shown in Figure 15, is one of the novel aspects of the proposed study due to its unique dimensions and shape.

The given inaccuracy of adaptation may be that it transitions from an unbalanced to a balanced medium may inversely affect the pattern of the proposed antenna. To address this problem, it would be helpful to use a balanced–unbalanced transformer at the end of the cable, which can provide the necessary isolation to prevent common-mode currents or other methods as given in [26]. For future activity, we plan to implement one or more of these solutions to the proposed antenna in order to supply more benefits to UWB applications, specifically in medicine.

**Author Contributions:** Conceptualization, H.N.A. and Y.I.A.; methodology, H.N.A., Y.I.A.; software, H.N.A.; validation, M.K., L.D. and Y.I.A.; formal analysis, H.N.A., Y.I.A. and F.F.M.; investigation, H.N.A.; resources, M.B.; data curation, H.N.A.; writing—original draft preparation, H.N.A. and Y.I.A.; writing—review and editing, M.B., E.U. and F.F.M.; visualization H.N.A.; supervision, M.K. and L.D.; project administration, M.K. and H.L.; funding acquisition, H.L. All authors have read and agreed to the published version of the manuscript.

**Funding:** This work was supported by the National Key Research and Development Program of China (Grant no. 2017YFA0204600), the National Natural Science Foundation of China (Grant no. 51802352) and the Fundamental Research Funds for the Central Universities of Central South University (Grant no. 2018ZZTS355).

**Acknowledgments:** The author would like to thanks Central South University for technical support.

**Conflicts of Interest:** The authors declare no conflict of interest.

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
