# Peer review of "Bandwidth Improvement in Bow-Tie Microstrip Antennas: The Effect of Substrate Type and Design Dimensions"

_applsci, doi:10.3390/app10020504_

Round 1

Reviewer 1 Report

The paper "Bandwidth improvement in bow tie microstrip antennas: The effect of substrate type and design dimensions" by Awl et al. reports a sound study of bow tie microstrip antennas in the frequency range from 4-8 GHz.

The paper is well written and the explanations and arguments are sound.

In the pdf version of the paper Figure 9 is not displayed correctly (not on the same page as the figure caption).

I think this paper will be of interest for the microwave community and I support publishing.

Author Response

Comments and Suggestions for Authors

In the pdf version of the paper Figure 9 is not displayed correctly (not on the same page as the figure caption). I think this paper will be of interest to the microwave community and I support publishing.

Answer: Thank you very much for your suggestion. In the revised manuscript, the figure has been inset correctly

Reviewer 2 Report

In this reviewer's opinion this article has little scientific sound to be published in a journal. talk about bandwidth improvement but it is not understood if it is relative to the author's own antennas.

It is more or less obvious that in this type of antenna the substrate is a necessary evil so if its thickness tends to zero better.

There are numerous sentences that do not bring scientific information and should be avoided; Here is an example:"The designed antenna was fabricated, tested and the measured results were compared with the simulation results. In the first attempt of investigation, we were unable to achieve a precise result."

The authors report that they use two microstrip lines when they actually use a parallel line transmission line. Part of the reported inaccuracy of adaptation may be that it transitions from an unbalanced to a balanced medium without any kind of transformation. apparently did not model this transition in the simulation.

Author Response

Comments and Suggestions for Authors

In this reviewer's opinion, this article has little scientific sound to be published in a journal. talk about bandwidth improvement but it is not understood if it is relative to the author's own antennas.

It is more or less obvious that in this type of antenna the substrate is a necessary evil so if its thickness tends to zero better.

Answer: Thank you for your comment It is achieved from the result of the paper that the smaller thickness provides wider bandwidth with lower gain. Therefore, the thickness can be selected based on the application. It means the substrate thickness can be used to control obtained gain and bandwidth.

There are numerous sentences that do not bring scientific information and should be avoided; Here is an example: "The designed antenna was fabricated, tested and the measured results were compared with the simulation results. In the first attempt of investigation, we were unable to achieve a precise result."

Answer: Thank you for your attention. The above sentences have been removed from the revised manuscript.

The authors report that they use two microstrip lines when they actually use a parallel line transmission line. Part of the reported inaccuracy of adaptation may be that it transitions from an unbalanced to a balanced medium without any kind of transformation. apparently did not model this transition in the simulation.

Answer :

Thank you for your comment. According to the comments of the Reviewer #2, we have added reference 22 and 23 to the manuscript and revise the conclusion section as below

In order to utilize bow-tie type UWB antennas Kim et al in [22] designed a balun which is integrated into the antenna. (in the introduction section )

The given inaccuracy of adaptation may be that it transitions from an unbalanced to a balanced medium may inversely affect the pattern of the proposed antenna. To address this problem, it would be helpful to use a balanced – unbalanced transformer at the end of the cable which provides necessary isolation to prevent common-mode currents or other methods given in [23]. As a future activity, we plan to implement one or more of these solutions to the proposed antenna in order to supply more benefits in UWB applications specifically in medicine. (in the conclusion section)

Reviewer 3 Report

In this contribution authors investigated the effect of substrate properties (permittivity, thickness and dimension) with the aim to widen the bow tie antenna bandwidth. the study is based on two different design of bow tie antenna and validated by experimental results.

The presented study and analysis are interesting and helpful for engineers and researchers. However the novelty is not clear. The strategies based on modified the bow tie antenna geometry to improuve the antenna bandwidth are not news.

In my opinion, the paper is not clear enough (novelty) and not well organized which makes it difficult to read and understand,

1-Figure 2 concerns the antenna geometry, while in the corresponding coments there are Figure 2a (line 127) and figure 2b (line 130) !!!

2- Figure 3: same title like Figure 2, without any information !!!

3- Figure 4 and Figure 5 show the same fabricated antenna (with different thicknesses) and can be grouped in one figure.

4- Figures 9 and 11 are not well placed by creating the pdf file (correction of this error with a new pdf generation are needed)

Author Response

Comments and Suggestions for Authors

In this contribution, the authors investigated the effect of substrate properties (permittivity, thickness, and dimension) with the aim to widen the bow-tie antenna bandwidth. the study is based on two different designs of the bow-tie antenna and validated by experimental results.

The presented study and analysis are interesting and helpful for engineers and researchers. However, novelty is not clear. The strategies based on modified the bow-tie antenna geometry to improve the antenna bandwidth are not news.

In my opinion, the paper is not clear enough (novelty) and not well organized which makes it difficult to read and understand,

Answer: Thank you very much for your comments: We think that this work can be used as a practical guide which helps researchers to view substrate selection as a method to control antenna performance (gain, bandwidth, polarization ). This point can be considered as the novelty of this work which may not be seen before.

We tried to reorganize the paper, especially by the help of your comments, to be more understandable  

1-Figure 2 concerns the antenna geometry, while in the corresponding comments there is Figure 2a (line 127) and figure 2b (line 130) !!!

Answer: Thank you for your attention. The figures now well embedded in the revised manuscript

2- Figure 3: the same title as Figure 2, without any information !!!

Answer: Thanks again. The title of figure 3 modified to be distinguished with figure 2.

3- Figure 4 and Figure 5 show the same fabricated antenna (with different thicknesses) and can be grouped in one figure.

Answer: Thank you for your comment, figure 4 and figure 5 are grouped in one figure.

4- Figures 9 and 11 are not well placed by creating the pdf file (correction of this error with a new pdf generation are needed)

Answer: Thanks for your attention. The new pdf file has been created and double-checked for the errors.

Round 2

Reviewer 2 Report

The authors argue as novelty that “the thickness can be selected based on the application. It means the substrate thickness can be used to control gain and bandwidth ”. People who design antennas know that substrate characteristics influence the properties of the antenna, so substrate selection is always carefully done with regard to the intended properties of the antenna. The author's conclusions do not add anything new.

The article has several flaws but two of them are unacceptable in a good quality magazine:

1- Characterizing a balanced antenna with an unbalanced power is incorrect and it is not enough to mention an article where there is a balun to undo the error: it is compulsory to add the corresponding balun to the antenna so that it can be truly characterized. It is not enough to say that it will be a future activity.

2-Antenna radiation diagrams (measured) must be made and presented.

Reviewer 3 Report

The suggested corrections have been made and the report is well written organised. This paper can be accepted for publication in the Applied Sciences journal

Author Response

Thank you so much. Your comments and suggestions indeed made our paper batter not only in terms of scientific aspects but also in terms of technical aspects as well in the first round.